# Reduced Levels of Cerebrospinal Fluid/Plasma Aβ40 as an Early Biomarker for Cerebral Amyloid Angiopathy in RTg-DI Rats

**DOI:** 10.3390/ijms21010303

**Published:** 2020-01-01

**Authors:** Xiaoyue Zhu, Feng Xu, Michael D. Hoos, Hedok Lee, Helene Benveniste, William E. Van Nostrand

**Affiliations:** 1George & Anne Ryan Institute for Neuroscience, Department of Biomedical and Pharmaceutical Sciences, University of Rhode Island, Kingston, RI 02881, USA; xiaoyuezhu123@uri.edu (X.Z.); feng_xu@uri.edu (F.X.); mhoos@enzolifesciences.com (M.D.H.); 2Enzo Life Sciences, 10 Executive Blvd, Farmingdale, NY 11735, USA; 3Department of Anesthesiology, Yale University, New Haven, CT 06520, USA; hedoklee@gmail.com (H.L.); helene.benveniste@yale.edu (H.B.)

**Keywords:** cerebral amyloid angiopathy, amyloid β-protein, transgenic rat, cerebrospinal fluid, biomarker

## Abstract

The accumulation of fibrillar amyloid β-protein (Aβ) in blood vessels of the brain, the condition known as cerebral amyloid angiopathy (CAA), is a common small vessel disease that promotes cognitive impairment and is strongly associated with Alzheimer’s disease. Presently, the clinical diagnosis of this condition relies on neuroimaging markers largely associated with cerebral macro/microbleeds. However, these are markers of late-stage disease detected after extensive cerebral vascular amyloid accumulation has become chronic. Recently, we generated a novel transgenic rat model of CAA (rTg-DI) that recapitulates multiple aspects of human CAA disease with the progressive accumulation of cerebral vascular amyloid, largely composed of Aβ40, and the consistent emergence of subsequent microbleeds. Here, we investigated the levels of Aβ40 in the cerebrospinal fluid (CSF) and plasma of rTg-DI rats as CAA progressed from inception to late stage disease. The levels of Aβ40 in CSF and plasma precipitously dropped at the early onset of CAA accumulation at three months of age and continued to decrease with the progression of disease. Notably, the reduction in CSF/plasma Aβ40 levels preceded the emergence of cerebral microbleeds, which first occurred at about six months of age, as detected by *in vivo* magnetic resonance imaging and histological staining of brain tissue. These findings support the concept that reduced CSF/plasma levels of Aβ40 could serve as a biomarker for early stage CAA disease prior to the onset of cerebral microbleeds for future therapeutic intervention.

## 1. Introduction

Cerebral amyloid angiopathy (CAA) is a common cerebral small vessel disease that involves the accumulation of amyloid β-protein (Aβ) primarily in small- and medium-sized arteries and arterioles of the meninges and cerebral cortex as well as along the capillaries of the cerebral microvasculature [1,2,3,4]. CAA is found, to varying degrees, in nearly 80% of elderly individuals [5,6]. Since CAA results from cerebral vascular deposition of Aβ, it is not surprising that this condition commonly coexists in patients with Alzheimer’s disease (AD) [1,2]. However, independent of AD, clinically CAA is a significant contributor to vascular-mediated cognitive impairment and dementia (VCID) [3,4,7]. As a prominent small vessel disease CAA contributes to the cognitive decline in VCID in several ways by promoting perivascular neuroinflammation, impaired cerebral blood flow and ischemic infarcts, cerebral microbleeds, and larger hemorrhages, all of which can result in neuronal dysfunction, neuronal loss and white matter damage [3,8,9,10].

Presently, the clinical diagnosis of CAA primarily relies on the detection of cerebral microbleeds and larger hemorrhages by imaging techniques [10,11]. Previously, a set of criteria was established, known as the ‘Boston Criteria’, that determines a diagnosis of probable CAA based on the presence and anatomical location of cerebral microbleeds [11]. An updated version of the “Boston Criteria” was proposed incorporating additional key imaging biomarkers that detect other cerebral vascular injuries associated with amyloid containing vessels including cortical superficial siderosis, enlarged perivascular spaces and white matter hyperintensities [12,13,14]. Also, anatomical distribution patterns of amyloid accumulation in whole brain detected by positron emission tomography (PET) and Aβ specific radiotracer imaging studies has been informative [15,16]. Although these additional imaging markers have improved the clinical diagnosis of probable CAA, they largely represent signs of late-stage disease that emerge after the extensive accumulation of cerebral vascular amyloid has become chronic. Markers that detect the earlier stages of CAA, prior to these late-stage imaging indications, would be beneficial to monitor disease progression in patients while at the same time avoiding therapeutic interventions which might trigger cerebral hemorrhage such as thrombolytic agents [17,18]. Alternatively, biomarkers of early onset CAA disease would help to identify patients that could be enrolled in novel therapeutically directed clinical trials.

Recently, we generated a novel transgenic rat model of CAA designated rTg-DI that faithfully recapitulates many features of human capillary CAA type-1 [19]. The rTg-DI rats express human AβPP in neurons harboring the familial CAA Dutch E693Q/IowaD694N mutations and produce chimeric Dutch/Iowa mutant Aβ in brain. This model exhibits early-onset and progressive microvascular/capillary amyloid accumulation in many forebrain brain regions including the cortex, hippocampus and thalamus that is largely composed of the Aβ40 peptide. However, there is little accumulation of fibrillar amyloid in larger cerebral vessels in this model. The deposition of cerebral microvascular amyloid is accompanied by robust perivascular neuroinflammation and associated with regular microbleeds [19]. Accordingly, rTg-DI rats provide an invaluable preclinical platform to follow development of CAA pathologies.

In the present study, we show that rTg-DI rats exhibit consistent and progressive accumulation of primarily Aβ40 peptide in the form of microvascular CAA throughout the cortex, hippocampus and thalamus starting at about three months of age. In addition, rTg-DI rats develop consistent cerebral microbleeds that are readily detected by magnetic resonance imaging (MRI) beginning at six months of age with expansion to twelve months of age and confirmed by histological analysis of perivascular hemosiderin deposits at these ages. Measurements of cerebrospinal fluid (CSF) Aβ40 in one month old rTg-DI rats, that do not yet exhibit CAA, represent the homeostatic levels of this peptide prior to disease onset. However, at three months of age, when rTg-DI rats start exhibiting CAA but do not yet show microbleeds, the CSF levels of Aβ40 precipitously and uniformly dropped. At six months of age and later, with further accumulation of CAA and emergence of microbleeds in the brain, CSF levels of Aβ40 continued to drop. Parallel analysis of rTg-DI plasma Aβ40 showed similar trends with lower levels at the onset of CAA at three months and further reductions at six to twelve months. These findings strongly indicate that a reduction in Aβ40 levels in biological fluids represent an early disease related biomarker that correlates with the onset of CAA prior to the emergence of subsequent cerebral microbleeds that are detected by neuroimaging and histological confirmation. Finally, this study underscores the translational utility of rTg-DI rats as a valid preclinical model to further develop biomarkers and a platform to test therapeutic interventions for CAA.

## 2. Results

### 2.1. rTg-DI Rats Show Early-Onset and Progressive Accumulation of Cerebral Vascular Amyloid

rTg-DI rats were designed to specifically express human AβPP harboring the adjacent Dutch E693Q and Iowa D694N familial CAA mutations in neurons in the brain [19]. The accumulation of chimeric Dutch E22Q/Iowa D23N CAA mutant human Aβ in rTg-DI rats over a period of twelve months was measured by ELISA analysis of soluble and insoluble brain fractions (Figure 1). The levels of soluble Aβ40 were very low at one month of age but rose over three to twelve months of age. In contrast, soluble Aβ42 levels were extremely low at one month and remained low through twelve months. On the other hand, the levels of insoluble Aβ40 were relatively low at one month but dramatically increased from three to twelve months. In contrast, the levels of insoluble Aβ42 were ≤ 10% the levels of insoluble Aβ40 at each age. These findings indicate that the accumulation of Aβ peptides in rTg-DI rats is very low at one month, but markedly increases as the rats age to twelve months and is primarily composed of Aβ40.

We recently reported that rTg-DI rats develop the progressive accumulation of cerebral microvascular amyloid that is largely composed of Aβ40 [19]. The ELISA data show low levels of cerebral Aβ peptides at one month (Figure 1). Similarly, at one month of age there was no evidence of microvascular CAA present in rTg-DI rats (Figure 2A). However, from three to twelve months there was the emergence and dramatic increase in the amount of microvascular CAA that paralleled the striking increase of cerebral (soluble and insoluble) Aβ40 peptide (Figure 2B–E). Together, these findings indicate that elevated cerebral Aβ40 levels and microvascular CAA levels both emerge at about three months of age and dramatically increase over the course of twelve months in rTg-DI rats.

### 2.2. rTg-DI Rats Develop Consistent Thalamic Microbleeds Detected by MRI and Histology

We longitudinally evaluated the emergence of cerebral microbleeds as detected by T2* mapping by MRI from the onset of microvascular CAA at three months to late-stage disease at nine months of age. In rTg-DI rats, the quantitative T2* maps (Figure 3C) clearly displayed bilateral thalamic ‘bleeds’ as represented by distinct areas with very low T2* values (T2* ≤ 20 ms) at nine months (noted by black arrows), consistent with previous findings reported by *ex vivo* MRI [19]. Notably, wild-type rats showed no evidence of microbleeds over the course of this study (Figure 3B,D). Thalamic microbleeds were consistently detected in the four rTg-DI rats on T2* maps as early as six months of age (Figure 3D,E). The quantitative T2* maps allowed for assessment of microbleed volume changes over time in rTg-DI rats and this analysis revealed emergence of small microbleeds (∼2 mm^3^) at six months of age with up to three-fold volume expansion as the animals aged from six to nine months (Figure 3D,E).

To support the MRI findings presented in Figure 3, we performed quantitative histological evaluation for perivascular hemosiderin deposits for further confirmation of thalamic cerebral microbleeds in rTg-DI rats as they aged from one to twelve months. In one month old rTg-DI rats, prior to microvascular CAA deposition, no hemosiderin deposits were detected in the thalamus (Figure 4A). Similarly, at three months of age, when microvascular CAA appears, there is still no histological evidence for cerebral microbleeds (Figure 4B), consistent with the MR imaging data presented in Figure 3. However, at six months of age, with more extensive CAA, thalamic perivascular hemosiderin deposits are evident (Figure 4C,E). The extent of hemosiderin deposition increases sharply at twelve months of age (Figure 4D,E), again highly consistent with the imaging findings. Together, these findings clearly show that cerebral microbleeds do not develop in rTg-DI rats until several months after the onset of cerebral microvascular amyloid deposition.

### 2.3. CSF and Plasma Levels of Aβ40 Markedly Drop at the Inception of Microvascular CAA in rTg-DI Rats

Previously, it was reported that the levels of Aβ40, the major isoform of Aβ found in CAA deposits, are reduced in probable CAA patients diagnosed by the presence of cerebral microbleeds [20,21,22]. Therefore, we performed cross sectional measurements of Aβ40 in the CSF of cohorts of rTg-DI rats as they progressed through different stages of disease to determine the trajectory of this marker. At one month, prior to the accumulation of appreciable cerebral Aβ40 or evidence of any microvascular CAA, the mean CSF levels of Aβ40 were ≈5500 pg/mL (Figure 5). At three months, with the marked accumulation of cerebral Aβ40 and the initial formation of CAA, there was a precipitous ≈70% drop in CSF Aβ40 levels to ≈ 1750 pg/mL (*p* < 0.0001). As rTg-DI rats aged further to six and twelve months, with more extensive cerebral Aβ40 and CAA accumulation, coupled with the emergence and progression of cerebral microbleeds, there were further declines in CSF Aβ40 levels to ≈18% and ≈10% the initial levels observed in one month old rTg-DI rats. Parallel measurements of CSF Aβ42 levels, although much lower than Aβ40 levels, showed similar declines with the onset and progression of microvascular CAA (data not shown).

We next extended the ELISA analysis for Aβ40 to plasma samples collected from the different aged rTg-DI rats (Figure 6). At one month, prior to the onset of microvascular CAA, the plasma levels of Aβ40 were ≈60-fold lower than what was measured in CSF and tended to show somewhat more variation. Nevertheless, at three months, with the onset of microvascular CAA, there was a significant ≈54% drop in the plasma levels of Aβ40 (*p* < 0.02). The levels of plasma Aβ40 continued to drop at six months of age to ≈12% of the initial one-month plasma levels and, in this case, appeared to level off to twelve months of age. Aβ42 levels in plasma appear to be extremely low and were not detected in the ELISA. Together, these findings show that both CSF and plasma levels of Aβ40 in rTg-DI rats are relatively high prior to the onset of CAA but sharply decline when cerebral microvascular amyloid begins to deposit. This acute decrease in CSF and plasma Aβ40 occurred early and precedes the emergence of cerebral microbleeds, but continues to decline as the animals age further and the disease severity progresses.

## 3. Discussion

CAA is a common cerebral small vessel disease of the elderly and a prominent comorbidity of AD that promotes and exacerbates VCID, yet our ability to diagnose CAA remains limited to late-stage neuroimaging markers for this condition. The neuroimaging markers used in the ‘Boston Criteria’ diagnosis is centered around vascular and perivascular changes that are associated with cerebral blood vessels impacted by the chronic and extensive accumulation of cerebral vascular amyloid. The major scoring factors for clinical diagnosis of CAA is the lobar presence of a large macrobleed, which occurs in very severe cases of disease, or the presence of multiple lobar microbleeds, which is more common and observed in chronic, late-stage disease with extensive cerebral vascular amyloid [11]. Additional neuroimaging markers that aid clinical diagnosis include perivascular changes that occur in the vicinity around CAA-affected vessels. For example, cortical superficial siderosis indicates perivascular iron accumulation in pial arteries and arterioles reflecting cerebral microbleeds [12,14,23]. Another noted perivascular change associated with CAA is the presence of dilated perivascular spaces, particularly around penetrating arterioles with amyloid deposition [8,24]. Although the cause of this perivascular alteration remains unclear this space contains CSF and interstitial fluid (ISF) that under normal conditions plays an important role in Aβ clearance from brain [25,26,27,28]. The presence of amyloid in these vessels promotes local perivascular inflammation and may disrupt normal CSF flow and exchange with ISF leading to enlargement of these spaces [4,29]. Although these neuroimaging markers are useful in diagnosing chronic, late-stage disease with extensive CAA, they are not helpful in detecting early stages of emerging cerebral vascular amyloid accumulation that could be more amenable to therapeutic interventional strategies and allow for clearance of the emerging perivascular Aβ deposits.

Parenchymal amyloid plaques commonly found in AD primarily contain the longer Aβ42 species. Previous studies have shown that as brain parenchymal plaques accumulate in AD there is a significant decrease in the levels of Aβ42 detected in CSF as this peptide accumulates in brain [30,31]. Accordingly, CSF Aβ42 levels have provided a surrogate biomarker for the parenchymal plaque burden in brain and have assisted in the clinical diagnosis of AD [32,33]. In contrast to plaques, CAA deposits are largely composed of the shorter Aβ40 peptide [1,2]. This suggests that a decrease in CSF Aβ40 levels could reflect the presence and burden of CAA. Indeed, studies have shown that patients diagnosed with CAA, based on late-stage neuroimaging biomarkers for cerebral microbleeds and cortical superficial siderosis, presented with lower levels of CSF Aβ40 [20,34]. However, the utility of CSF Aβ40 levels to serve as a potential biomarker for earlier stage CAA, prior to the onset of microbleeds and other perivascular changes, remains unclear.

Most animal models to study CAA have largely involved the use of transgenic mouse lines that express human AβPP generally producing highly elevated levels of Aβ peptides in brain with or without familial CAA mutations [35,36,37,38]. Although these mouse models have been helpful in studying the pathogenesis of CAA their usefulness in modeling the human disease has been met with limitations including the small size of the mouse brain, which has hampered neuroimaging studies, and the variable presentation of cerebral microbleeds. Our recently generated transgenic rat model rTg-DI more faithfully recapitulates many of the pathological features of human small vessel CAA including early-onset and progressive accumulation of cerebral microvascular fibrillar amyloid, perivascular neuroinflammation, consistent and robust development of cerebral microhemorrhages and small vessel occlusions and behavioral deficits [19]. The rTg-DI rats used in this study provided a unique opportunity to evaluate the trajectories of Aβ40 in CSF and plasma in cohorts of animals that consistently progressed from the early-onset to late-stage disease pathologies. Indeed, from the accumulation of Aβ peptides in brain, to the development of CAA, to the emergence and expansion of microbleeds and to measurements of CSF/plasma Aβ40 the cohorts rTg-DI rats progressed through each stage of disease with uniformity.

The CSF compartment is an important exchange reservoir with the ISF compartment that surrounds the cellular components of the brain. This dynamic interaction plays an important clearance route for Aβ and other cellular metabolic waste products through glymphatic system transport and/or alternate intramural perivascular drainage pathways [25,26,27,28]. Thus, the CSF compartment can reflect the ongoing clearance of Aβ and provide a snapshot of Aβ pathology and burden in the brain. For example, the chronic pathological accumulation of Aβ42 in parenchymal plaques in AD brain is reflected by decreased levels of this specific Aβ isoform in CSF and is associated with a decline in cognitive function [32,33]. Similarly, our findings demonstrate that the selective accumulation of Aβ40 in CAA deposits is reflected by early onset decreased levels of this Aβ species in CSF.

Another important route for Aβ clearance from the CNS is across the blood-brain barrier of the cerebral endothelium into blood for peripheral removal [39,40]. This clearance route is facilitated by endothelial Aβ transporters including low density lipoprotein receptor related protein 1 (LRP1) and p-glycoprotein [39,41,42]. However, the levels of Aβ peptides in plasma are much lower than in CSF and can be more variable and, therefore, is less reliable in serving as a biomarker for AD [43,44]. In the case of using rTg-DI rats these shortcomings are further compounded by the finding that compared with non-mutated wild-type Aβ the chimeric Dutch/Iowa CAA mutant Aβ poorly binds LRP1 and is much less effectively transported across the blood-brain barrier into the circulation [45]. Indeed, the level of Aβ40 detected in plasma of rTg-DI rats was < 2% of what was measured in the CSF. Nevertheless, we still found that Aβ40 levels in plasma of rTg-DI rats dramatically drop with the onset of cerebral microvascular amyloid accumulation, thus mirroring what was observed in the CSF.

We recently reported that in CAA deposits in transgenic mice and humans, as well as in rTg-DI rats, the amyloid fibrils adopt a distinct anti-parallel configuration [19,46]. The present findings suggest that once Aβ40 forms fibrillar amyloid deposits in cerebral microvessels and capillaries, this may act as a nidus for further seeding of additional soluble Aβ40 to expand vascular fibrillar amyloid with this distinct anti-parallel conformation and further impair its clearance through either the CSF or into the peripheral circulation.

The trajectory of disease markers in the rTg-DI model of microvascular CAA are summarized in Figure 7. In rTg-DI rats the accumulation of Aβ peptides in brain and the onset of CAA began early, at about two to three months of age, and dramatically increased over twelve months. The emergence of cerebral microbleeds, confirmed by *in vivo* MR imaging and histological evaluation, occur significantly later starting at around six months, with further expansion at nine and twelve months. The present findings show that at the onset of CAA, much earlier than the emergence of cerebral microbleeds, there is a precipitous drop in the CSF/plasma levels of Aβ40, the chief component of CAA deposits. The reductions in CSF Aβ40 levels in rTg-DI rats are consistent with prior studies using CSF collected from probable CAA patients with MRI confirmed cerebral microbleeds [20,21,22,34]. However, the present findings show that reductions in CSF/plasma Aβ40 occur much earlier with the onset of CAA and prior to microbleeds. This underscores the value of this novel model to identify other possible biomarkers that correlate with disease state.

## 4. Materials and Methods

### 4.1. rTg-DI Rats

All animal experiments were approved by the local Institutional Animal Care and Use Committees at the University of Rhode Island (project #AN1718-008; approval dates 12/11/2017-12/10/2020) and Yale University (project #2019-20132; approval dates 11/1/2019-10/31/2022) and conducted in accordance with the United States Public Health Service’s Policy on Humane Care and Use of Laboratory Animals. rTg-DI rats were designed to express human AβPP (isoform 770) harboring the Swedish K670N/M671L, Dutch E693Q, and Iowa D694N mutations in neurons under control of the Thy1.2 promoter and produce chimeric Dutch/Iowa CAA mutant Aβ in their brains [19]. Transgenic offspring were determined by PCR analysis of tail DNA. All subsequent analyses were performed with heterozygous transgenic rats.

### 4.2. CSF Collection

CSF was collected from the cisterna magna of rTg-DI rats at designated ages. Rats were deeply anesthetized with inhalation of isoflurane and then mounted on a stereotaxic unit. A midline incision was made beginning between the ear and ending approximately 2.5 cm caudally. The fascia was retracted and muscles dissected, exposing the atlanto-occipital membrane. Using a #11 scalpel, a small slit was made along the midline of the membrane and underlying dura under a surgical microscope. The CSF was collected through the dura slit by using a fine tip pipette and aliquoted into sterile Eppendorf tube and frozen at −80 °C. Approximately 150 µL of CSF was collected from each rat.

### 4.3. Plasma Isolation

Rat blood was collected by terminal cardiac puncture from anesthetized rTg-DI rats at the designated ages. Blood was collected in one tenth volume of 3.8% sodium citrate to prevent coagulation. Blood was centrifuged at 8000× *g* for 5 min at room temperature to remove platelets and cellular components. Plasma samples were stored at −80 °C until ELISA analysis.

### 4.4. Brain Tissue Collection and Preparation

Rats were euthanized at designated time points and perfused with cold-PBS, forebrains were removed and dissected through the mid-sagittal plane. One hemisphere was immersion-fixed with 70% ethanol overnight and subjected to increasing sequential dehydration in ethanol, followed by xylene treatment and embedding in paraffin. Alternatively, brains were fixed with 4% paraformaldehyde overnight at 4 °C and subjected to increasing concentrations (10%, 20%, 30%) of sucrose in PBS, then embedded in OCT compound (Sakura Finetek Inc., Torrance, CA, USA) and snap-frozen in dry ice. Other hemispheres were collected, frozen on dry ice and stored at –80 °C. Sagittal sections were cut at 10 µm thickness using a Leica RM2135 microtome (Leica Microsystems Inc., Bannockburn, IL, USA), placed in a flotation water bath at 40 °C, and then mounted on Colorfrost/Plus slides (ThermoFisher Scientific, Houston, TX, USA). In some cases, coronal sections were cut at 20 μm thickness from frozen brains using a Leica CM1900 cryostat (Leica Microsystems Inc.), stored in PBS with 0.02% sodium azide at 4 °C.

### 4.5. ELISA Quantitation of Aβ Peptides

The levels of soluble and insoluble Aβ40 and Aβ42 were determined by performing specific ELISAs as described [47,48]. Briefly, brain hemispheres that were flash frozen and pulverized in liquid nitrogen. A soluble fraction was obtained by homogenizing tissue with 10 µL/mg of 1M sodium carbonate, 500 mM NaCl, pH 11.5 and 0.5 mm zirconium oxide beads in a bullet blender. Aliquots were spun at 1600× *g* at 4 °C for 20 min. The supernatant was removed, which was the soluble fraction. The remaining pellet was suspended in 5 M guanidine-HCl, 50 mM Tris, pH 8.0 and rotated at room temperature for 3 h. Samples were centrifuged and the supernatant was collected, which was the insoluble fraction. For each of the two fractions, a sandwich ELISA was performed. Antibody reagents for the Aβ ELISAs were generously provided by Lilly Research Laboratories, Indianapolis, IN, USA. In the sandwich ELISAs Aβ40 and Aβ42 peptides were captured using the carboxyl-terminal specific antibodies m2G3 and m21F12, respectively, and biotinylated m3D6, specific for the N-terminus of human Aβ, was used for detection followed by streptavidin-HRP (Amdex RPN4401V; Fisher Scientific, Pittsburgh, PA, USA). Plates were developed using KPL SureBlue (SeraCare, Milford, MA, USA) and read using a Spectramax M2 plate reader (Molecular Devices, Sunnyvale, CA, USA). Each sample lysate was measured in triplicate and compared to linear standard curves generated with known concentrations of human Aβ. The same ELISA format was used to measure soluble Aβ levels in CSF and plasma collected from each rat.

### 4.6. Immunohistochemical Analysis

Antigen retrieval was performed by treating the tissue sections with proteinase K (0.2 mg/mL) for 10 min at 22 °C. Primary rabbit polyclonal antibody to collagen type IV was used to visualize cerebral microvessels (1:100; ThermoFisher, Rockford, IL, USA). Primary antibody was detected with Alexa Fluor 594-conjugated donkey anti-rabbit secondary antibody (1:1000). Staining for fibrillar amyloid was performed using thioflavin S. Prussian blue iron staining was performed to detect hemosiderin deposits reflecting signs of previous microhemorrhage.

### 4.7. Quantitative Histological Analysis of CAA Load and Microbleeds

The percent area amyloid coverage of cerebral microvessels and percent area iron staining in the thalamic region was determined in rats at each of the specified ages using stereological principles as previously described [19,36].

### 4.8. Magnetic Resonance Imaging Analysis

For non-invasive MRI imaging, rTg-DI rats (*N* = 4) and age-matched WT rats (*N* = 4) were lightly anesthetized with dexmedetomidine (0.015 mg/kg/h) and low dose isoflurane 0.5–1% as previously described [49]. MRI imaging was performed on a Bruker 9.4T MRI and images of the rat brain were acquired using a 40 mm volume transmit and receive coil. The 3D gradient echo imaging parameters were acquired with the following parameters: TR/TE/FA = 60 ms/2~32/15°, NEX = 6, resolution= 0.23 × 0.23 × 0.23 mm, scan time= 50 min. A proton density weighted anatomical MRI of each rat’s brain was acquired at the same spatial resolution. Following MRI, the anesthesia was discontinued, and the rats allowed to recover. Rats were scanned at three, six, and nine months of age.

Quantitative 3D T2* maps were calculated from the 3D multiple gradient echo (MGE TE = 2~32 ms) MRIs by assuming mono-exponential relationship between the signal and TEs [50]. Deposition of paramagnetic containing blood product, such as ferritin, and T2* values have been reported to be linearly correlated. T2* values ≤ 20 ms within the thalamus was identified on the parametric T2* maps in each rat using the Amira software segmentation editor (Amira 6.4, ThermoFisher Scientific, Houston, TX, USA). The number of voxel with T2* ≤ 20 ms was converted into mm^3^ and used as an estimate of ‘total hemorrhagic load’ in the thalamus.

### 4.9. Statistical Analysis

Histological and biochemical data were analyzed by *t*-test at the 0.05 significance level.

## Figures and Tables

**Figure 1 ijms-21-00303-f001:**
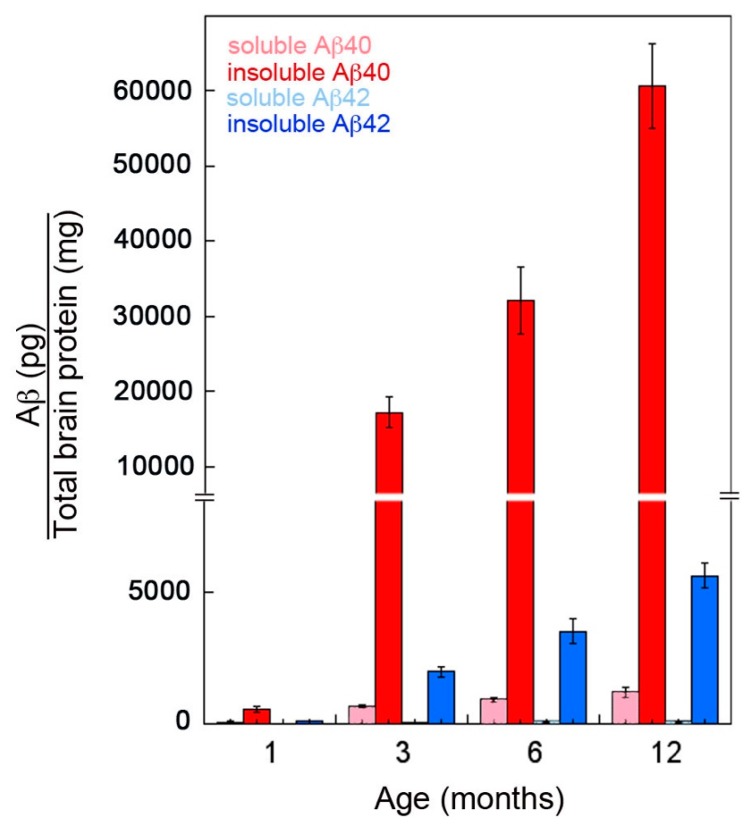
Progressive accumulation of Aβ peptides in rTg-DI rats. The levels of soluble and insoluble Aβ40 and Aβ42 peptides in the forebrain of rTg-DI rats aged from one to twelve months were measured by ELISA as described in “Methods”. The data presented are the means ± S.D. of triplicate measurements performed in 5–6 rTg-DI rats per age group.

**Figure 2 ijms-21-00303-f002:**
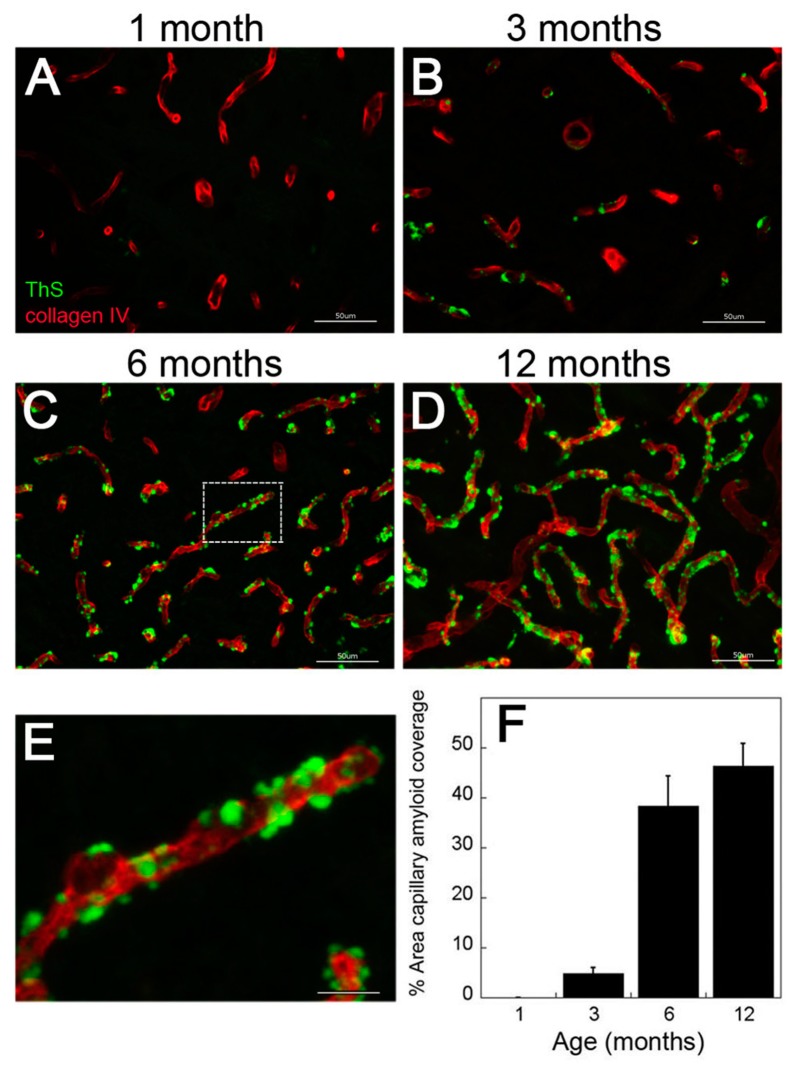
Progressive accumulation of cerebral microvascular amyloid in rTg-DI rats. Representative brain sections showing the thalamic region from rTg-DI rats aged to one month (**A**), three months (**B**), six months (**C**) and twelve months (**D**) that were stained for fibrillar amyloid using thioflavin-S (green) and immunolabeled for collagen type IV to identify cerebral microvessels (red). Scale bars = 50 µM. (**E**) Enlarged inset of panel **C** showing capillary localization of fibrillar amyloid. Scale bar = 10 µm. (**F**) Quantitation of microvascular thioflavin-S positive amyloid load in the thalamic region of rTg-DI rats aged one to twelve months. Data shown are mean ± S.D. of 5–6 rTg-DI rats per group.

**Figure 3 ijms-21-00303-f003:**
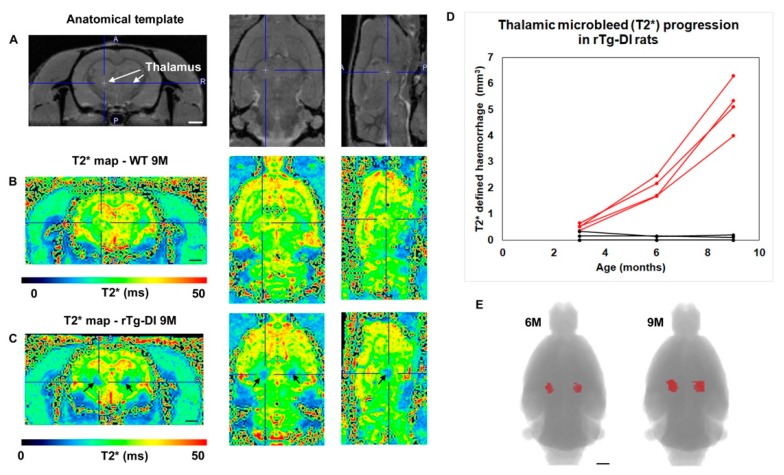
Microbleed progression in thalamus of rTg-DI rats over nine months visualized by MRI. Female rTg-DI CAA rats and wild-type rats were scanned longitudinally at 3, 6 and 9 months of age to track development of microbleeds as defined by T2* parametric mapping in rTg-DI rats. (**A**) Proton density weighted (PDW) anatomical MRIs presented in three orthogonal planes from a 3-month old rTg-DI CAA for demonstrating the position of the thalamus (white arrows) where the presence of microbleeds is typically noted. (**B**) T2* parametric, color coded images of the brain presented in three orthogonal planes from a 9 months of age wild type rat. The ‘blue’ and ‘red’ colors represent low and high T2* values, respectively. (**C**) T2* brain map from a 9 M old rTg-DI rat in the same orthogonal planes as in B, with black arrows pointing towards large, dark blue areas in the thalamus representing low T2* (≤20 ms) values indicating the presence of ferritin (hemorrhage). Note that the location of the thalamic microbleeds is symmetrical. (**D**) Quantitative assessment of thalamic microhemorrhage progression over time as defined by T2* ≤ 20 ms from four different rTg-DI CAA rats (red) in comparison to four wild-type rats (black). Small hemorrhages start emerging at 6 months (but not at 3 months) and continue to expand to nine months (on average a three-fold increase in volume). (**E**) 3D volume rendering of the microbleed in a rTg-DI rat based on T2* ≤ 20 ms, showing the expansion of the microhemorrhage area over time. Scale bars = 3 mm.

**Figure 4 ijms-21-00303-f004:**
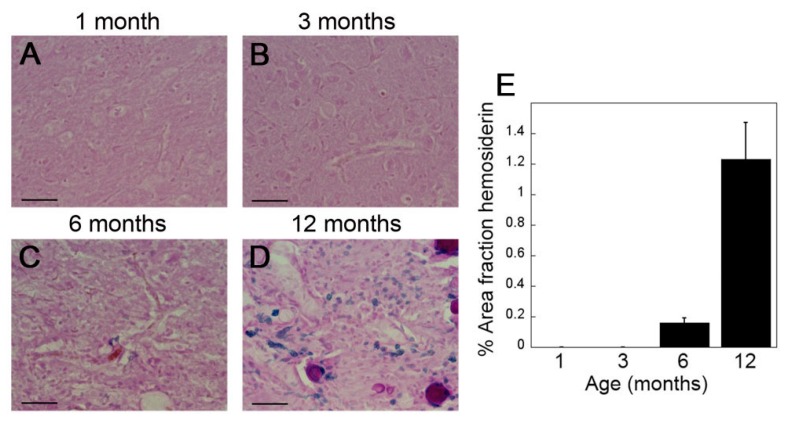
Emergence of cerebral microbleeds in rTg-DI rats. Representative brain sections showing the thalamic region from rTg-DI rats aged to one month (**A**), three months (**B**), six months (**C**) and twelve months (**D**) that were stained for hemosiderin to identify microhemorrhages (blue). Scale bars = 50 µm. (**E**) The percent area fraction of hemosiderin staining was quantitated in the thalamus of 1, 3, 6, and 12 months old rTg-DI rats. Data represent the mean ± S.D. of 6–7 rTg-DI rats per group.

**Figure 5 ijms-21-00303-f005:**
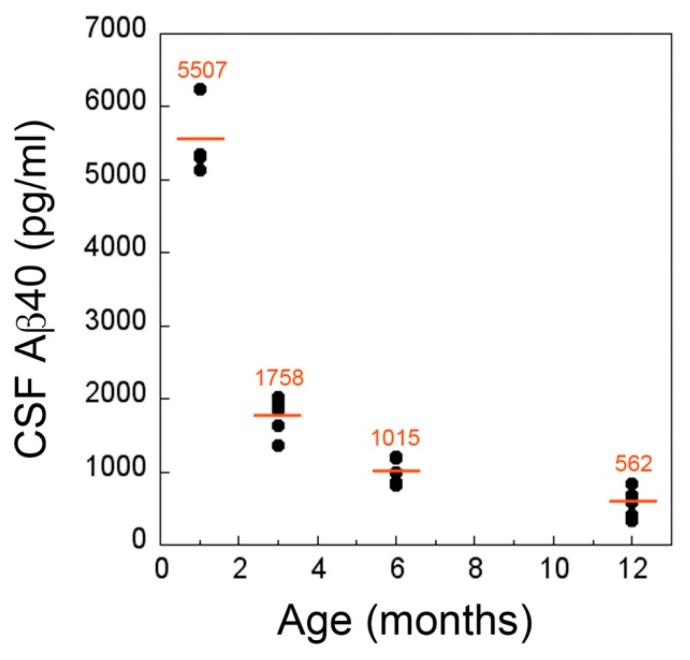
Decreased levels of Aβ40 in CSF of rTg-DI rats at the inception and progression of microvascular CAA. The levels of Aβ40 in CSF collected from rTg-DI rats as they aged from one to twelve months was determined by ELISA. The data show the mean CSF Aβ40 levels at each time point from *n* = 5–6 rTg-DI rats per group. At three months of age, when CAA first appears, there is precipitous drop in CSF Aβ40 levels that continue to decline with progressing CAA.

**Figure 6 ijms-21-00303-f006:**
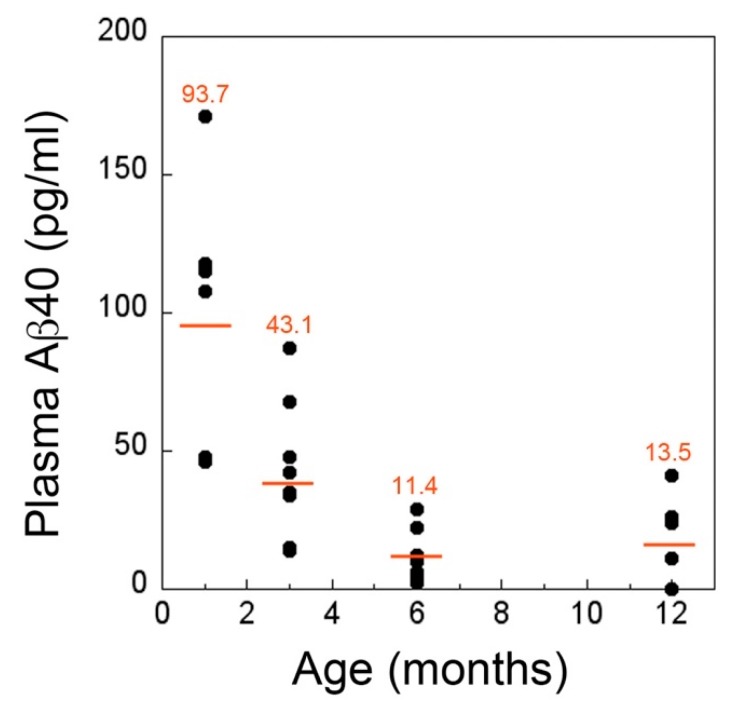
Decreased levels of Aβ40 in plasma of rTg-DI rats at the inception and progression of microvascular CAA. The levels of Aβ40 in plasma collected from rTg-DI rats as they aged from one to twelve months was determined by ELISA. The data show the mean plasma Aβ40 levels at each time point from *n* = 6–8 rTg-DI rats per group. At three months of age, when CAA first appears, there is marked reduction in plasma Aβ40 levels that continue to decline with progressing CAA.

**Figure 7 ijms-21-00303-f007:**
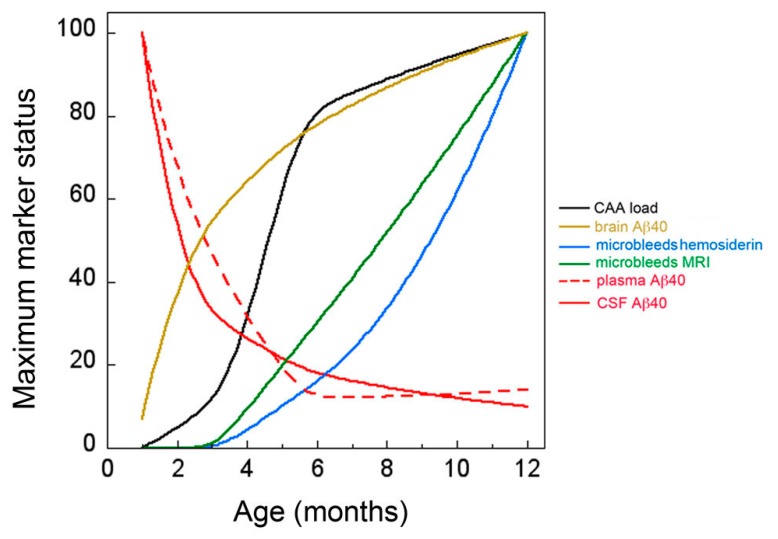
Trajectory of CSF and plasma Aβ40 levels with the progression of CAA and associated microbleed pathology in rTg-DI rats. At one month of age, the levels of Aβ40 in CSF and plasma are relatively high, whereas little Aβ40 has accumulated in brain tissue, and CAA and microbleeds are not present. At three months of age, the levels of Aβ40 in brain sharply increase and CAA begins to develop. At this same time, there is dramatic decrease in the levels of Aβ40 in both CSF and plasma. At six months of age, the levels of Aβ40 in brain and CAA continue to increase and microbleeds begin to emerge. The levels of Aβ40 in both CSF and plasma continue to decline. Finally, at twelve months of age, there is a continued increase in brain Aβ40, CAA load and expansion of cerebral microbleeds that is accompanied by further reduction in CSF Aβ40.

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
