# Peer review of "Reduced Levels of Cerebrospinal Fluid/Plasma Aβ40 as an Early Biomarker for Cerebral Amyloid Angiopathy in RTg-DI Rats"

_ijms, 2020, doi:10.3390/ijms21010303_

Round 1
Reviewer 1 Report
MDPI, International Journal of Molecular Sciences
Article: ijms-671771: Reduced levels of cerebrospinal fluid/plasma Aβ40 as an early biomarker for cerebral amyloid angiopathy in rTg-DI rats.
Authors: Xiaoyue Zhu, Feng Xu, Michael D. Hoos, Hedok Lee, Helene Benveniste & William E. Van Nostrand.
General Comments:
The manuscript by Zhu and colleagues followed the levels of the brain, plasma and cerebrospinal fluid levels of Aβ40 over a 12 month period in transgenic cerebral amyloid angiopathy (CAA) producing rTg-DI rats. Therein, Aβ40 concentrations in rat forebrain were markedly elevated at 3 months, with levels of the amyloidogenic peptide increasing thereafter to 12 months. Histological analyses revealed concomitantly increased ThS-reactive amyloid in the microvasculature and microbleeds in the perivascular space as identified by hemosiderin deposition. Similarly, MRI revealed thalamic microbleed progression that was readily identified at 6 months, thereby proceeding vascular Aβ40 deposition. Interestingly, Aβ40 levels in plasma and most notably in CSF were found to markedly decrease at 3 months of age, coincident with the progression of microvascular CAA in rTg-DI rats. Taken collectively, this study highlights the use of Aβ40 as an early biomarker of CAA progression with relevance to Alzheimer’s disease (AD) and vascular mediated cognitive impairment and dementia (VCID) seen in humans.
Comments:
Abstract, Page 1, Line 31: The beta symbol is missing in Aβ40.
Introduction, Page 2, Lines 22 – 23: Could the authors be more specific in relation to the localisation of Aβ in the brain of their transgenic rTg-DI rat model?
Line 36: This sentence is confusing: “do not yet microbleeds” and would benefit from being reworded.
Line 40: Should this not read as ‘a reduction in Aβ40 levels’.
Results, Page 3, Line 6: ‘markedly increases out’ is a strange term, what is meant here?
Line 8, Figure 1: How do these values relate to wild type (WT) rats? Such data is not depicted in this study, nor can it be readily assessed from the group’s previous publication [19]. Have you measured Aβ40/42 in WT animals in plasma and CSF? Presumably, these levels are greatly elevated in the rTg-DI versus WT animals?
Line 19: Thioflavin S (ThS) binds to the common beat-pleated sheet structural motif of mature amyloid fibrils of both Aβ40 and Aβ42. Therefore, could the authors comment upon the predominance of Aβ40 being identified by this method in the vasculature of rTg-DI rats?
Page 4, Line 1, Figure 2: Figures depicting fluorescence micrographs are especially small. Magnified inserts would help to demonstrate the co-localisation of ThS-reactive amyloid on collagen type IV-positive cerebral microvessels.
Line 9, Figure 3: The position of the thalamus (Fig. 3A) should also be noted in the text and not just in the figure legend.
Line 10: Change to ‘9 months of age’.
Figure 3: It would be helpful to have a legend for Fig. 3D to highlight the data for rTg-DI rats versus the wild type.
Page 5, Line 7: Remove ‘a’ so the text reads as ‘pointing toward large, dark blue areas’.
Figure 4: There is either an issue here with the Figure, or more notably it's figure legend. No data is shown for (G) and the colours of the bars in the histogram do not correspond with those stated in the legend (i.e. grey versus red).
Page 6, Line 5: Do these rats develop other neuropathologies related to Alzheimer’s disease (AD) - i.e. senile plaques of beta amyloid / neurofibrillary tangles of hyperphosphorylated tau protein? Did you try staining the brain tissues of these animals for related neuroinflammation or these aforementioned neuropathological hallmarks?
Discussion, Page 8, Line 19: Was there any evidence for amyloid plaque formation in these brain tissues of rTg-DI rats co-incident with the increasing levels of Abeta42, as noted in rat forebrains?
Lines 43 - 47: Sentence confusing in its current form. Needs rewording for clarity (i.e. fibrillar amyloid accumulation in microvessels and capillaries may act as a nidus for Aβ seeding and hence its precipitation and deposition in the vasculature).
Page 9, Line 12: Remove ”the” before “Figure 7”.
Materials and Methods, Page 10, Line 30: Reference [47] is difficult to access. A brief outline of the method would be beneficial here.
Author Response
RESPONSE TO REVIEWER #1
Abstract, Page 1, Line 31: The beta symbol is missing in Aβ40.
Response: Corrected
Introduction, Page 2, Lines 22 – 23: Could the authors be more specific in relation to the localisation of Aβ in the brain of their transgenic rTg-DI rat model?
Response: We expanded this sentence to “This model exhibits early-onset and progressive microvascular amyloid accumulation in many forebrain brain regions including the cortex, hippocampus and thalamus that is largely composed of the Ab40 peptide” to further describe the localization of vascular Ab deposition.
Line 36: This sentence is confusing: “do not yet microbleeds” and would benefit from being reworded.
Response: The word “show” was missing to this sentence and has now been added.
Line 40: Should this not read as ‘a reduction in Aβ40 levels’.
Response: We corrected this and added ‘a’ to this statement.
Results, Page 3, Line 6: ‘markedly increases out’ is a strange term, what is meant here?
Response: We corrected this sentence to better state “These findings indicate that accumulation of Ab peptides in rTg-DI rats is very low at one month but markedly increases as the rats age to twelve months and is primarily composed of Ab40.”
Line 8, Figure 1: How do these values relate to wild type (WT) rats? Such data is not depicted in this study, nor can it be readily assessed from the group’s previous publication [19]. Have you measured Aβ40/42 in WT animals in plasma and CSF? Presumably, these levels are greatly elevated in the rTg-DI versus WT animals?
Response: The measures performed in our ELISAs were specific for human Ab peptides. We did not perform measures for rodent Ab peptides that would require different antibodies. However, in this transgenic model the level of human AbPP expression is low and below the level of endogenous rat AbPP expression [19]. Therefore, presumably, the amount of Ab peptides generated from the human AbPP transgene expression would be near physiological levels of rat Ab as well. However, due to the effects of the familial CAA Dutch/Iowa mutations in Ab on increased fibrillogenesis and less efficient clearance of the peptide this leads to the large accumulation of this mutant Ab peptide in the microvessels of these transgenic rats.
Line 19: Thioflavin S (ThS) binds to the common beat-pleated sheet structural motif of mature amyloid fibrils of both Aβ40 and Aβ42. Therefore, could the authors comment upon the predominance of Aβ40 being identified by this method in the vasculature of rTg-DI rats?
Response: The reviewer is correct about ThS binding to both Ab40 and Ab42 fibrils. However, in our earlier work [19] we clearly demonstrated that the microvascular fibrillar amyloid in rTg-DI rats is largely (>90%) composed of the shorter Ab40. On lines 13-14 we added this point “We recently reported that rTg-DI rats develop progressive accumulation of cerebral microvascular amyloid that is largely composed of Ab40 [19].”
Page 4, Line 1, Figure 2: Figures depicting fluorescence micrographs are especially small. Magnified inserts would help to demonstrate the co-localisation of ThS-reactive amyloid on collagen type IV-positive cerebral microvessels.
Response: As requested by the reviewer we have included new fluorescence micrographs with an additional magnified inset that clearly show the co-localization of fibrillar amyloid with capillary/microvessels.
Line 9, Figure 3: The position of the thalamus (Fig. 3A) should also be noted in the text and not just in the figure legend.
Response: As requested in the text related to Figure 3 we noted that the thalamic regions designated by black arrows.
Line 10: Change to ‘9 months of age’.
Response: As requested, changed to ‘9 months of age’.
Figure 3: It would be helpful to have a legend for Fig. 3D to highlight the data for rTg-DI rats versus the wild type.
Response: This comment is unclear as there is a legend for Fig. 3D in the Figure 3 legend.
Page 5, Line 7: Remove ‘a’ so the text reads as ‘pointing toward large, dark blue areas’.
Response: As requested, ‘a’ was removed.
Figure 4: There is either an issue here with the Figure, or more notably it's figure legend. No data is shown for (G) and the colours of the bars in the histogram do not correspond with those stated in the legend (i.e. grey versus red).
Response: The figure legend has been corrected as pointed out by the reviewer.
Page 6, Line 5: Do these rats develop other neuropathologies related to Alzheimer’s disease (AD) - i.e. senile plaques of beta amyloid / neurofibrillary tangles of hyperphosphorylated tau protein? Did you try staining the brain tissues of these animals for related neuroinflammation or these aforementioned neuropathological hallmarks?
Response: The rTg-DI rats do not develop parenchymal fibrillar amyloid plaques. This is consistent with the Dutch and Iowa CAA disorders in human. We have performed studies characterizing the neuroinflammatory aspects associated with CAA in this model. However, those studies are the topic of another manuscript that is currently under revision at a different journal.
Discussion, Page 8, Line 19: Was there any evidence for amyloid plaque formation in these brain tissues of rTg-DI rats co-incident with the increasing levels of Abeta42, as noted in rat forebrains?
Response: As stated above, rTg-DI rats do not develop parenchymal fibrillar amyloid plaques. This is consistent with the Dutch and Iowa CAA disorders in human. As the rTg-DI rats age there is indeed some accumulation of Ab42, although at much lower levels than Ab40. However, even at 12 months of age parenchymal amyloid plaques are not evident in these rats.
Lines 43 - 47: Sentence confusing in its current form. Needs rewording for clarity (i.e. fibrillar amyloid accumulation in microvessels and capillaries may act as a nidus for Aβ seeding and hence its precipitation and deposition in the vasculature).
Response: As requested by the reviewer we have clarified this statement to read “The present findings suggest that once Ab40 forms fibrillar amyloid deposits in cerebral microvessels and capillaries this may act as a nidus for further seeding of additional soluble Ab40 to expand vascular fibrillar amyloid and further impair its clearance through either the CSF or into the peripheral circulation.
Page 9, Line 12: Remove ”the” before “Figure 7”.
Response: “the” was removed.
Materials and Methods, Page 10, Line 30: Reference [47] is difficult to access. A brief outline of the method would be beneficial here.
Response: As requested by the reviewer we have now provided details for the Ab ELISAs in ‘Materials and Methods’ on pages 10-11 of the revised manuscript.
Reviewer 2 Report
I like this paper very much. It provides novel insights into the mechanisms surrounding Abeta40 deposition around capillaries. Couple of minor points:
Differences from human disease should be noted and explained, if possible, like the absence of arteriolar/larger vessel deposition What was the promoter used for the transgene? Was it restricted to cerebral expression? Although previously published in ref 19, I do not have this paper at hand. the mechanism of capillary microbleeds should be described in more detail, if possible, based on the histological evidence involvement of vessels in white matter is becoming of increasing importance, given the human occurrence of WMH. Some details of this in the rat model would be of interest.Author Response
RESPONSE TO REVIEWER #2
Differences from human disease should be noted and explained, if possible, like the absence of arteriolar/larger vessel deposition.
Response: In the Introduction in lines 20-21 we more clearly state that rTg-DI rats recapitulate the features of human capillary CAA type-1. Further, in lines 25-26 we also state that there is little accumulation of fibrillar amyloid in larger cerebral vessels in this model.
What was the promoter used for the transgene?
Response: In Materials and Methods in lines 6-8 we clarify that the human AbPP transgene is expressed in neurons under control of the Thy1.2 promoter.
Was it restricted to cerebral expression? Although previously published in ref 19, I do not have this paper at hand.
Response: As stated above, human AbPP transgene was expressed in neurons.
The mechanism of capillary microbleeds should be described in more detail, if possible, based on the histological evidence involvement of vessels in white matter is becoming of increasing importance, given the human occurrence of WMH. Some details of this in the rat model would be of interest.
Response: The mechanisms responsible for microbleeds in this model or in humans with CAA is not clear although it has been hypothesized that it may involve breakdown of vessel basement membranes, loss of vessel elasticity, focal increases in vascular tension or activation of proteolytic enzymes. The rTg-DI rats now provide a model to further investigate these possible mechanisms. Regarding WM damage, in a separate study currently under review at another journal we have documented with neuroimaging and histopathology robust WM loss and damage in rTg-DI rats as they age with progressive CAA, which occurs independent of appreciable vascular amyloid accumulation in the WM itself.